# Clinical Signs in 166 Beagles with Different Genotypes of Lafora

**DOI:** 10.3390/genes15010122

**Published:** 2024-01-19

**Authors:** Thomas Flegel, Christine Dirauf, Alexandra Kehl, Josephine Dietzel, Annette Holtdirk, Ines Langbein-Detsch, Elisabeth Müller

**Affiliations:** 1Department for Small Animals, Veterinary Faculty, Leipzig University, 04103 Leipzig, Germany; josephine.dietzel@kleintierklinik.uni-leipzig.de; 2Department of Molecular Biology, Laboklin GmbH & Co. KG, 97688 Bad Kissingen, Germany; dirauf@laboklin.com (C.D.); kehl@laboklin.com (A.K.); langbein@laboklin.com (I.L.-D.); mueller@laboklin.com (E.M.); 3RQM+ (Germany) GmbH, 59229 Ahlen, Germany; a.holtdirk@cro-kottmann.de

**Keywords:** NHLRC1, myoclonic, seizure, photosensitivity

## Abstract

Lafora disease (LD) is a genetic disease affecting beagles, resulting in seizures in combination with other signs. The aim of this study was to describe the clinical signs of LD in beagles with different NHLRC1 genotypes. One hundred and sixty-six beagles were tested for an NHLRC1 gene defect: L/L (*n* = 67), N/L (*n* = 32), N/N (*n* = 67). Owners were asked to participate in a survey about the clinical signs of LD in their dogs. These were recorded for the three possible genotypes in the two age groups, <6 years and ≥6 years. In all genotypes, nearly all the signs of LD were described. In the age group ≥ 6 years, however, they were significantly more frequent in beagles with the L/L genotype. If the following three clinical signs occur together in a beagle ≥ 6 years—jerking of the head, photosensitivity and forgetting things he/she used to be able to do—98.2% of these dogs are correctly assigned to the L/L genotype. If one or two of these signs are missing, the correct classification decreases to 92.1% and 13.2%, respectively. Only the combination of certain signs truly indicates the L/L genotype. Yet, for many dogs, only genetic testing will provide confirmation of the disease.

## 1. Introduction

Lafora is a rare neurological storage disorder caused by an autosomal recessive genetic defect and mainly manifests as myoclonic epileptic seizures [1,2]. The disease was first described in humans in 1911, but was later found in cattle, fennec foxes, cockatiels, cats and dogs [3,4,5,6,7,8]. In affected humans, two causative genes have been identified, the epilepsy, progressive myoclonus type 2A and 2B (EPM2A and EPM2B) genes, which encode the two proteins, laforin (protein tyrosine phosphatase) and malin (ubiquitin ligase) [9]. The absence of either protein results in the formation of poorly branched, hyperphosphorylated glycogen, which accumulates into Lafora bodies and, in this altered form, is stored in all brain regions [9,10]. The exact mechanism of how this accumulation causes the typical myoclonic and generalized tonic–clonic seizures is not known [9]. 

Lafora disease has been described in the following dog breeds: basset hound, beagle, Chihuahua, French bulldog, Newfoundland, pointer, Welsh corgi, miniature poodle and miniature dachshund [11,12,13,14,15,16,17,18,19,20,21,22,23,24,25,26,27,28]. The beagle seems to be the most frequently affected breed in Germany (personal communication Kehl A, Labogen). In beagles, a repetitive sequence of 12 nucleotides in the NHL Repeat Containing E3 Ubiquitin Protein Ligase 1 (NHLRC1; EPM2B) gene, which has an autosomal recessive inheritance pattern, has been shown to cause Lafora disease [20]. The clinical signs of the disease in beagles have been described in a few individual case reports and small series as well as in a recent, more extensive study [13,14,20,24,26,27]. Common clinical signs include spontaneous myoclonic epilepsies triggered by visual or auditory stimuli, generalized tonic–clonic epileptic seizures, impaired vision and hearing, as well as mental retardation and abnormal behavior, with onset between 6 and 13 years of age [13,14,20,24,26,27]. A multicenter study of 28 beagles diagnosed with Lafora disease included one dog that, despite only being heterozygous for the defect in the NHLRC1 gene, showed typical signs of the disease, contrary to what would be expected given the assumed autosomal recessive inheritance [27]. 

This led to the aim of this study which was to describe the phenotypic expression of the different NHLRC1 genotypes in a larger group of beagles and to investigate whether some symptoms or a combination of multiple symptoms are more common in affected animals than others to gain a more accurate picture of the disease. Based on this, it was analysed whether the clinical signs can be used to draw conclusions about the Lafora genotype.

## 2. Materials and Methods

The database of a commercial laboratory for genetic testing in animals (Laboklin GmbH & Co. KG, Bad Kissingen, Germany) was searched for beagles that were tested between February 2018 and December 2020 for the known genetic defect in the NHLRC1 gene. The animals were included regardless of the genotype. Genetic testing was performed via fragment length polymorphism analysis as previously described [28]. Genomic DNA was isolated from EDTA blood using the MagNA Pure system (Roche Deutschland Holding GmbH, Mannheim, Germany). The sequence including the dodecamer repeat expansion was amplified by PCR using primers NHLRC1-F (5′-aggtgtgcttcgagaggttc-3′) and NHLRC1-R (5′-cccccttctctccaaactg-3′). Investigation of the presence/absence of the expansion was carried out by gel electrophoresis. PCR product length of about 430 bp is expected in wildtype dogs; a length between 620 bp and 700 bp is expected in homozygous mutant dogs. 

However, only animals whose EDTA blood samples were submitted by veterinarians from Germany, Austria and Switzerland were included. The submitting veterinarians were contacted by email with a request to contact the dog owners and ask them to participate in an online survey about the clinical signs in their dogs. If the animal owners agreed, they were then sent the survey link (Wordpress.com survey tool, ExpressTech, Bangalore, India).

The survey consisted of the following components: questions on signalment (age, sex), questions on the reasons for genetic testing (breeding reasons, dog showing clinical signs suggestive of Lafora, general interest), questions on clinical signs (Table 1). The survey was structured in a way that owners could stop answering and continue later if they were not sure about their answers.

For the evaluation of the data, the dogs were divided into two age groups: those younger than 6 years at the time of the survey, which were not expected to show any signs of Lafora disease based on previous publications, and those that were 6 years or older and thus at the age at which Lafora disease causes clinical signs [13,14,20,24,26,27].

Statistical analyses were carried out using SPSS for Windows, version 24.0 (SPSS Inc., IBM, Armonk, NY, USA). Categorical and nominal data are provided as absolute or relative frequencies and were analyzed using the chi-square test and Fisher’s exact test, respectively. The dependence of a dichotomous variable on other independent influencing factors (predictors) was examined using the binary logistic regression model. For all tests performed, a two-sided significance test was carried out and a *p*-value < 0.05 was assumed to be statistically significant for all statistical tests.

## 3. Results

In the relevant period, 622 beagles that had been tested for Lafora were identified in the database. Of these, 557 samples were requested by veterinarians. Then, 326 veterinarians or veterinary institutions were contacted with a request to participate in the study. Overall, a total of 166 beagles were included in the study this way. The total response rate in relation to the number of beagles which could potentially be included was therefore 29.8%. The distribution of the three genotypes resulting from these patients, homozygous for the Lafora mutation Lafora/Lafora (L/L), heterozygous for the Lafora mutation normal/Lafora (N/L) and free of the Lafora mutation normal/normal (N/N), as well as the corresponding sex distribution are summarized in Table 2. It cannot be excluded that some of the 67 animals with the L/L genotype were already part of a previous study in which clinical signs were described in 27 beagles with this genotype alone [27]. The mean age of the beagles was 4.0 years in the age group < 6 years and 10.1 years in the age group ≥ 6 years.

Table 3 and Table 4 summarize the occurrence of clinical signs in dogs with the three different genotypes. The odds ratio, which is also included, indicates how much the risk of having an L/L genotype increases when a particular clinical sign is present. Figure 1 shows the age at which the first epileptic seizure occurred in dogs with the three different genotypes of Lafora. There was no difference in how often any of the clinical signs examined occurred in dogs with the N/N and N/L genotypes, nor in the group < 6 years of age, or in the group ≥ 6 years of age.

Performing a multiple logistic regression analysis shows that 83.1% of the variability between the genotypes N/N plus N/L and L/L can be explained by the following 10 parameters: age group (<6 years; ≥6 years), generalized tonic–clonic epileptic seizures, jerking of the head, photosensitivity, seeking attention, coordination problems, difficulties climbing stairs, impaired hearing, has forgotten things he/she used to be able to do and panic attacks. By reducing the influencing factors to three (jerking of the head, photosensitivity, has forgotten things he/she used to be able to do), the number of cases that could be included increased from *n* = 132 to *n* = 156. The results of the multiple logistic regression analysis if those three factors are used are summarized in Table 5 and Table 6. If one of the three signs is missing, the correct assignment to the L/L genotype ranges from 92.1% to 95.9%. If only one of the three signs is present, the correct assignment drops to 13.2% to 68.9%, depending on which sign remains. Overall, the correct prediction of L/L or N/N plus N/L for beagles ≥ 6 years of age using the three factors: jerking of the head, photosensitivity, has forgotten things he/she used to be able to do, is 92.1%. 

## 4. Discussion

The aim of the study was to describe the phenotypic expression of the different NHLRC1 genotypes in a larger group of beagles, and based on this, to analyze whether the Lafora genotype can be inferred from the clinical signs. It has been shown that the combination of the three signs jerking of the head, photosensitivity and forgetting things he/she used to be able to do allow for the patient to be assigned to the L/L genotype with a probability of 99.4% if the patient is at least 6 years old at the time of testing. However, the absence of one or even more of these three signs does not completely rule out that the L/L genotype is present. The L/L genotype was always compared with the genotypes N/L and N/N as a whole, since a clinical distinction between the latter two genotypes could not be derived from the data in this study. It is therefore possible to predict the L/L genotype with a high degree of probability in beagles which are at least 6 years of age, but it cannot be deduced whether they are only heterozygous carriers of the genetic defect. Thus, for the final determination of the genotype, a DNA test will always be required.

In an earlier study, it was noticed that the typical signs of Lafora disease were also present in a heterozygous beagle [27]. This was surprising, as it was not expected that a heterozygous patient would show clinical signs of the disease, given the autosomal recessive inheritance [1,2]. The question was, therefore, raised as to whether additional, previously unknown genetic or environmental factors could influence the clinical expression of Lafora disease, as had already been discussed in the case of an affected human family [29]. And indeed, in the present study, such typical signs of Lafora were also described in beagles with the N/L genotype. Yet, the same was also true for beagles with the N/N genotype, even in the group under 6 years of age, when the disease in beagles is not yet clinically expressed [13,14,20,24,26,27]. 

A major weakness of this study is that concomitant diseases were not captured by the questionnaire. It is possible that some clinical signs observed by the owners were caused by other orthopedic or neurological diseases. Specifically, restless pacing, staring into space, lack of coordination, reduced vision, especially when seen in combination with seizures, can be, amongst others, manifestations of intracranial neoplasia, which are frequently seen in older dogs [30]. Structural epilepsy is the major differential diagnosis for dogs developing seizures after 5 years of age [31]. In addition, difficulties going up stairs and coordination problems could have been caused by chronic spinal cord diseases such as Hansen type II intervertebral disc disease or peripheral nerves system pathologies. That could have specifically influenced the owners’ observations in Beagles in the age group above 6 years. Another weakness of the present study is that the assessment of clinical signs was only based on the subjective opinion of the animal owners. It has to be assumed that most of the animal owners were aware of the clinical signs of Lafora disease, since they had intentionally tested their dog. Knowledge about the disease may well have influenced their perception of the signs and symptoms. One reason why this is particularly likely is that there was no difference in how the typical signs of Lafora were perceived in dogs with the N/L and N/N genotypes. As such, it may be difficult for an animal owner to distinguish between the presence of Lafora disease and an asymptomatic carrier or a dog that is free of the genetic defect only by looking at individual signs. However, an alternative explanation for the observation of clinical signs of Lafora disease by owners of dogs with the N/L and N/N genotype who were 6 years or older could be that some of the signs, such as forgetting things he/she used to be able to do, coordination disorders, loss of house training, decreasing playfulness, etc., result from normal ageing or are an expression of cognitive dysfunction which occurs in older age [32,33].

Similarly, the occurrence of generalized, tonic–clonic epileptic seizures is only of limited help in distinguishing between the three genotypes L/L, N/L and N/N. It is true that L/L animals which are at least 6 years of age suffer from epileptic seizures 6.5 times more often, but animals of the other two genotypes also suffer from these seizures (Table 4, Figure 1). There is, however, a difference between these genotypes regarding the age of onset of seizures. Dogs with the L/L genotype mainly seem to develop seizures at the age of six years with the onset of clinical signs of Lafora disease, whereas dogs with the N/N genotype may suffer from seizures as early as one year of age; but these seizures are more likely to be manifestations of idiopathic epilepsy, for which beagles are predisposed [34]. The few beagles with the L/L genotype that presented with generalized tonic–clonic epileptic seizures before 6 years of age may be animals that had developed idiopathic epilepsy at an early age and later developed additional signs of Lafora disease due to their genetic defect, although it cannot be completely ruled out that epileptic seizures are an early manifestation of Lafora disease and precede the other signs [32,33]. Alternatively, it has to be considered that clinical signs of Lafora in Beagles start much earlier than previously reported, but owners do not recognize those before they become more severe. The latter could be supported by the owner’s observation of several other clinical signs that were seen more frequently in a few beagles with the L/L genotype in comparison to N/N and L/N Beagles taken together in the age group ≤ 6 years such as: jerking of the head, photosensitivity, anxiousness, coordination problems, frequent blinking, impaired vision, difficulties learning new task and others.

In summary, it can be said that the combination of the three clinical signs jerking of the head, photosensitivity and forgetting things he/she used to be able to do allow the assignment of a patient to the L/L genotype with a probability of 99.4%, but that there is considerable overlap of the clinical signs in the three genotypes so a reliable assignment, especially of the N/N and N/L genotypes, always requires a genetic test.

## Figures and Tables

**Figure 1 genes-15-00122-f001:**
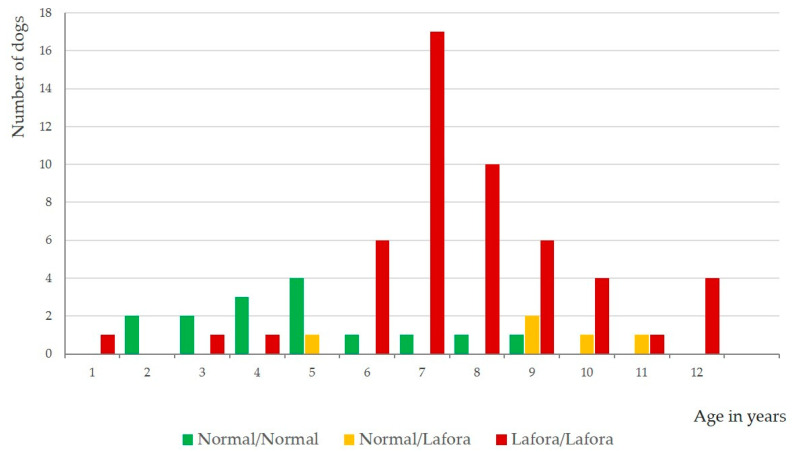
Age at which the first generalized tonic–clonic epileptic seizure occurred in beagles of the three different genotypes of Lafora (*n* = 71).

**Table 1 genes-15-00122-t001:** Questions on clinical signs associated with Lafora disease and response options which were provided to animal owners via online survey tool.

Question	Response Options
Does your dog show the following signs of disease? Generalized epileptic seizures: recurrent brief episodes (1–5 min) with several of the following signs: unconsciousness, falling over, rowing movements of the limbs, jaw smacking, salivation, uncontrolled urination or defaecation	yes; no; I cannot say
At what age did the signs mentioned above start?	1 to 14 at yearly intervals; 15 or older; I cannot say
How often, on average, does your dog experience the above-mentioned seizures?	every day; several times a weekonce a week; several times a month; once a month; 1–2 × every six months; once a year; highly variable frequency; I cannot say
Does your dog show the following signs of disease? jerking of the head when a hand moves towards the dog’s head, when the head is touched, when there is a sudden light or sound stimulus or even without a triggering stimulus	yes; no; I cannot say
At what age did the signs mentioned above start?	1 to 14 at yearly intervals; 15 or older; I cannot say
How often, on average, does your dog experience the above-mentioned jerks?	every day; several times a week; once a week; several times a month; once a month; 1–2 × every six months; once a year; highly variable frequency; I cannot say
Does your dog show the following behavioural changes (all the following abnormalities were asked separately)? - sensitivity to noise - photosensitivity - sleepiness - decreasing playfulness - aggression towards people - aggression towards animals - staring into space - restless pacing - panic attacks - anxiousness - seeking attention - loss of house training - has forgotten things he/she used to be able to do - difficulties learning new tasks - problems orientating in unfamiliar surroundings - frequent blinking - impaired hearing - impaired vision - difficulties climbing stairs - coordination problems (staggering gait)	yes; no; I cannot say
Is your dog still alive or has your dog unfortunately died or had to be put down?	he/she is still alive; he/she is no longer alive

**Table 2 genes-15-00122-t002:** Sex distribution among the three genotypes of beagles tested for the presence of the Lafora mutation differentiated into two age groups (*n* = 166).

	<6 Years of Age		≥6 Years of Age		Total
Genotype	Male	Female	Male	Female	
Lafora/Lafora	2	4	35	26	67
Normal/Lafora	4	11	9	8	32
Normal/Normal	14	22	16	15	67
total	20	37	60	49	166

**Table 3 genes-15-00122-t003:** Clinical signs in beagles < 6 years of age tested for Lafora and how their occurrence increases the risk (odds ratio) of having the L/L genotype compared to the combined group including N/L and N/N. The percentages indicate the frequency of the sign within a genotype (*n* = 57; N/N, genotype Normal/Normal; N/L, genotype Normal/Lafora; L/L, genotype Lafora/Lafora). Odds ratios are only provided for clinical signs with significant differences between groups. The basis for calculating percentages is the number “yes” or “no” responses for every clinical sign, whereas “I cannot say” responses were not included in this calculation.

Clinical Sign	Genotype			Odds Ratio for the Presence of the L/L Genotype	*p*-Value
	N/N	N/L	L/L		
generalized tonic–clonic epileptic seizures	3 (8.3%)	0 (0.0)	3 (50.0%)	16	0.012
jerking of the head	1 (2.9%)	0 (0%)	3 (50.0%)		0.003
sensitivity to noise	1 (2.8%)	2 (14.3%)	2 (33.3%)		0.084
photosensitivity	0 (0%)	0 (0%)	3 (50.0%)		0.001
anxiousness	3 (8.3%)	1 (7.7%)	4 (66.7%)		0.003
seeking attention	15 (42.9%)	6 (40.0%)	2 (33.3%)		1.000
coordination problems	0 (0%)	0 (0%)	4 (66.7%)		< 0.001
difficulties climbing stairs	0 (0%)	0 (0%)	3 (50.0%)		0.001
frequent blinking	2 (5.7%)	0 (0.0%)	4 (66.7%)		0.001
restless pacing	2 (5.6%)	0 (0%)	2 (33.3%)		0.051
staring into space	2 (5.6%)	0 (0%)	1 (16.7%)		0.298
impaired vision	1 (3.0%)	1 (6.7%)	2 (50.0%)		0.026
impaired hearing	2 (5.7%)	0 (0%)	3 (50.0%)		0.007
sleepiness	6 (16.7%)	0 (0%)	2 (33.3%)		0.200
decreasing playfulness	6 (16.7%)	1 (6.7%)	2 (33.3%)		0.237
loss of house training	1 (2.8%)	0 (0%)	1 (16.7%)		0.201
has forgotten things	0 (0%)	1 (6.7%)	1 (16.7%)		0.201
difficulties learning new tasks	0 (0%)	0 (0%)	2 (33.3%)		0.010
panic attacks	0 (0%)	0 (0%)	2 (3.3%)		0.010
problems in unfamiliar surroundings	0 (0%)	0 (0%)	3 (50.0%)		0.001
aggression towards people	0 (0%)	1 (0%)	1 (16.7%)		0.201
aggression towards animals	3 (8.3%)	2 (15.4%)	2 (33.3%)		0.163

**Table 4 genes-15-00122-t004:** Clinical signs in beagles ≥ 6 years of age tested for Lafora and how their occurrence increases the risk (odds ratio) of having the L/L genotype compared to the combined group including N/L and N/N. The percentages indicate the frequency of the sign within a genotype (*n* = 109; N/N, genotype Normal/Normal; N/L, genotype Normal/Lafora; L/L, genotype Lafora/Lafora). Odds ratios are only provided for clinical signs with significant differences between groups. The basis for calculating percentages is the number of “yes” or “no” responses for every clinical sign, whereas “I cannot say” responses were not included in this calculation.

Clinical Sign	Genotype			Odds Ratio for the Presence of the L/L Genotype	*p*-Value
	N/N	N/L	L/L		
generalized tonic–clonic epileptic seizures	12 (40.0%)	5 (29.4%)	48 (78.7%)	6.52	<0.001
jerking of the head	4 (13.8%)	3 (17.6%)	60 (98.4%)	334.3	<0.001
sensitivity to noise	6 (20.0%)	0 (0%)	37 (63.8%)	14.4	<0.001
photosensitivity	1 (3.6%)	2 (11.8%)	48 (81.4%)	61.1	<0.001
anxiousness	7 (24.1%)	3 (18.8%)	31 (55.4%)	4.3	0.001
seeking attention	15 (46.7%)	5 (29.4%)	44 (75.9%)	4.6	<0.001
coordination problems	5 (16.7%)	3 (17.6%)	38 (64.4%)	8.8	<0.001
difficulties climbing stairs	7 (23.3%)	3 (17.6%)	38 (54.8%)	7.4	<0.001
frequent blinking	2 (7.4%)	2 (11.8%)	31 (88.6%)	135	<0.001
restless pacing	6 (21.4%)	2 (12.5%)	27 (46.6%)	39	0.003
staring into space	8 (27.6%)	2 (13.3%)	32 (62.7%)	5.7	<0.001
impaired vision	3 (12.0%)	2 (13.3%)	29 (61.7%)	11.3	<0.001
impaired hearing	5 (18.5%)	1 (6.3%)	19 (40.4%)	4.2	0.009
sleepiness	6 (20.7%)	2 (12.5%)	37 (67.3%)	9.5	<0.001
decreasing playfulness	7 (23.3%)	2 (11.8%)	39 (70.9%)	10.3	<0.001
loss of house training	0 (0%)	1 (5.9%)	7 (11.5%)		0.134
has forgotten things	1 (3.3%)	0 (0%)	16 (27.1%)	17.1	<0.001
difficulties learning new tasks	0 (0%)	1 (6.7%)	15 (34.9%)	23.6	<0.001
panic attacks	0 (0%)	1 (6.3)	13 (22.4%)	13.0	0.003
problems in unfamiliar surroundings	1 (3.3%)	0 (0%)	18 (34.6%)	24.4	<0.001
aggression towards people	1 (3.3%)	0 (0%)	5 (8.2%)		0.234
aggression towards animals	2 (6.9%)	3 (18.8%)	9 (15.0%)		0.773

**Table 5 genes-15-00122-t005:** Results of the multiple logistic regression analysis to predict assignment to the genotype L/L or L/N + N/N using the combination of the three clinical signs provided in the table for all dogs together *n* = 156). The odds ratios refer to the odds of having the genotype L/L (*n* = 63).

Selection of Clinical Signs	Regression Coefficient	*p*-Value	Odds Ratio	Correct Assignment to Genotype L/L	Correct Assignment to Genotype N/L + N/N
jerking of the head	3.480	0.000	32.4	93.7%	91.4%
photosensitivity	2.222	0.008	9.2		
has forgotten things he/she used to be able to do	2.029	0.008	7.6		

**Table 6 genes-15-00122-t006:** Results of the multiple logistic regression analysis to predict assignment to the genotype L/L or L/N + N/N for dogs ≥ 6 years of age using the combination of the three clinical signs provided in the table (*n* = 101). The odds ratios refer to the odds of having the genotype L/L (*n* = 657).

Selection of Clinical Signs	Regression Coefficient	*p*-Value	Odds Ratio	Correct Assignment to Genotype L/L	Correct Assignment to Genotype N/L + N/N
jerking of the head	4.332	0.000	76.1	98.2%	84.1%
photosensitivity	1.979	0.023	7.2		
has forgotten things he/she used to be able to do	2.676	0.108	0.6		

## Data Availability

Data are available on request due to privacy restrictions.

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
