# Peer review of "Clinical Signs in 166 Beagles with Different Genotypes of Lafora"

_genes, 2024, doi:10.3390/genes15010122_

Round 1

Reviewer 1 Report

Comments and Suggestions for Authors

I appreciate the invitation to review this paper. My main background is in applied statistics with extensive experience in canine genomics. Broadly, the study describes Lafora symptoms to genotypic associations to NHLRC1 variants. The paper is well written with good an easy-to-follow arguments. The sample size is sufficient considering the small response rate of 29.8%.

The sub stratification of <6 and >6 years because it helps a lot in understanding in context the onset of the disease. This cutoff is based on an extensive preliminary evaluation of reported ages in the literature.

I only have a couple comments to improve the paper.

First, add the ORs and p-value columns to Table 3 just as in Table 4.

Second, the multivariate regression analysis described in lines 129-145 can be well presented as a table where the coefficients are shown. This will provide excellent context.

Last, please provide some details on the assay used to genotype the samples. I see you referenced the previous paper, but this having some of that would be helpful.

I recommend a minor revision to address these comments.

Reviewer 2 Report

Comments and Suggestions for Authors

Dear authors,

I have read your study and ask for revision of below mentioned issues.

LIN58 - The survey is lacking question on medical history of studied animals. These could potentially impact the recorded symptoms and discrepancies between genotypes and observed clinical signs.

LIN101 - The study you mention in the sentence, should be cited here.

Table 2 - Is there a reason why did you distinguish between neutered/spayed animals, if it is not mentioned anywhere else in the study? You should comment in the discussion if this could affect the results. 

Table 3 - The percentages differ throughout the columns for the same values (e.g. Genotype N/N - Jerking of head and Sensitivity to noise, both have only 1 recorded animal, but one has percentage 2.9, while the other 2.8). Occasion of this appear on several values throughout the entire table and needs to be fixed. 

Table 4 - Similar to the Table 3, the percentage values do not correspond to each other and to the total amount of recorded genotypes (e.g. Genotype N/L - Jerking of the head and Anxiousness, both have value 3 but percentages 17.6 and 18.8 respectively). Moreover, why are some of the values for Odds ratio missing (loss of house training, aggression towards people, aggression towards animals)?

LIN169-172 - Have the participants of the survey not been informed about the nature of the studied disease and the associated clinical signs? 

LIN191 - Some other neurological diseases (e.g. Wobbler disease, degenerative myelopathy) could be associated with some of the clinical signs that were reviewed in your studied. Those should probably also be discussed as they could explain some of the genotype discrepancies.

Round 2

Reviewer 2 Report

Comments and Suggestions for Authors

Dear authors,

thank you for addressing my questions regarding your study and correcting some of the errors.

I found no further issues with the paper.